# Population Density, Size Structure, and Reproductive Cycle of the Comestible Sea Urchin *Sphaerechinus granularis* (Echinodermata: Echinoidea) in the Pagasitikos Gulf (Aegean Sea)

**DOI:** 10.3390/ani10091506

**Published:** 2020-08-26

**Authors:** Dimitrios Vafidis, Chryssanthi Antoniadou, Vassiliki Ioannidi

**Affiliations:** 1Department of Ichthyology and Aquatic Environment, School of Agricultural Sciences, University of Thessaly, 38446 Volos, Greece; ioannidi.vasia@gmail.com; 2Department of Zoology, School of Biology, Aristotle University of Thessaloniki, 54124 Thessaloniki, Greece; antonch@bio.auth.gr

**Keywords:** population abundance, biometry, gonad-somatic index, histology, sea urchin, *Sphaerechinus*, Aegean Sea

## Abstract

**Simple Summary:**

Regular urchins are dominant grazers in the shallow sublittoral seabed. Several species are edible and commercially harvested, among which *Sphaerechinus granularis* is of increasing commercial importance due to the depletion of the common urchin *Paracentrotus lividus*. As there are very few studies on the biology of the species, the present work examines the population density, size structure, and reproduction of *S. granularis* in the Aegean Sea (eastern Mediterranean). Population density, in-situ estimated along transects, conforms to previous reports for the Aegean Sea. Size-structure, estimated from 20 randomly collected specimens at monthly base, showed one mode at 65–70 mm or 70–75 mm, according to the sampling location. Sea urchins had larger dimensions in the sheltered site. The monthly variation of the gonad-somatic index and the histological analyses of the gonads, estimated from the same 20 randomly collected specimens (per month and station), showed that the species reproduces once each year, in spring. The results of the present study provides baseline knowledge on the biology of *S. granularis* and are important for the viable management of its developing fishery.

**Abstract:**

*Sphaerechinus granularis* is a common grazer that lives in various sublittoral habitats, displaying typical covering behavior; i.e., putts shell-fragments, pebbles, and algae on its test. It is an edible species of increasing commercial importance due to the depletion of the common urchin’s, *Paracentrotus lividus*, stocks. Its biology, however, is not adequately studied over its distributional range. The present study examines population density, size structure, and reproductive biology of *S. granularis* in the Aegean Sea. Samplings were made with SCUBA-diving (8–10 m) and included: (i) visual census along transects to estimate density, and (ii) random collection of specimens at monthly intervals to assess biometry and gametogenesis. Population density had moderate values that almost doubled when inputted to Distance software. *S. granularis* had larger dimensions in the sheltered site; size-structures were unimodal (65–70 mm and 70–75 mm, in exposed and sheltered site, respectively). An annual reproductive cycle was evident, according to GSI and gonads’ histology, with a clear spawning peak in early spring. This pattern conforms to previous reports from the Atlantic, but precedes those from the Mediterranean (reproduction in summer). The provided baseline knowledge on the biology of *S. granularis* is important for the viable management of its developing fishery.

## 1. Introduction

The blunt or purple sea urchin, *Sphaerechinus granularis* (Lamarck 1816), is an edible species distributed in the NE Atlantic and the Mediterranean Sea [1]. The species lives in a variety of habitat types [2] in both hard [3] and soft bottoms covered by seagrass meadows [4] or on maërl beds [5]. Its bathymetric distribution expands from the intertidal down to 130 m depth [6], but it is most usually found in the shallow sublittoral (5 to 15 m) where it forms locally dense populations [7]. As other sea urchins, it manifests a cryptic behavior (Figure 1) by occupying crevices and by camouflaging, as it covers its test with shell fragments, pebbles, and algae [8].

The species is harvested for its gonads in France, Spain and in few other countries, since the beginning of 1980s, although it is less appreciated than *Paracentrotus lividus* (Lamarck, 1816), the target species in sea urchins fishery [5]. In recent years, however, the depletion of *P. lividus* natural stocks is shifting sea urchin fishery to alternative species such as *S. granularis*, which is a fast-growing, large-sized urchin that contains voluminous gonads [5]. In the Hellenic Seas, *S. granularis* is locally consumed, but it is not officially harvested, and so, the minimum landing size of 7 cm [9] is not applicable [7]. Unfortunately, fisheries data are completely lacking for the region.

Despite the increasing importance of *S. granularis* as a fishery resource, its biology is scarcely studied. More explicitly, the existing information is limited to the reproductive biology [1,5], and the life strategy of the species in relation to food availability [10] in the western Mediterranean and the NE Atlantic (Brittany). In Brittany, the species has a rather brief breeding period that begins in spring (April or May) and extends until early summer (June), in dependence to the annual thermal regime [5]. Seawater temperature affects mainly the onset of spawning, which exhibits inter-annual variability in relation to this environmental factor [9]. In Mediterranean, the breeding period begins in summer [1,11] but may extend until November [1]. It is widely accepted that the triggering factor initiating spawning is the rapid rise in temperature that usually occurs in spring in temperate seas [5,9,11], although differences in food quality are also involved [1]. For the Aegean Sea, there are no relevant studies other than a few reports on its distribution [12] and population ecology [7].

Within this context, the present work aims to study the population density, the size structure and the reproductive biology of *S. granularis* in the central part of the Aegean Sea (Pagasitikos Gulf).

## 2. Materials and Methods

### 2.1. Study Area and Field Sampling

The present study was carried out in Pagasitikos Gulf, central Aegean Sea, which is an enclosed, shallow water, meso-oligotrophic gulf [13]. Two stations were sampled: S1 (39°18′19.3″ N, 22°56′35.3″ E) and S2 (39°18′25.1″ N, 23°5′50.9″ E), located at the north and northeast parts of the Gulf, respectively (Figure 2). The sea bottom was sandy in both stations, either unvegetated or covered by moderately dense seagrass meadows, with large boulders and scattered rocks in the shallower zone (up to 5 m depth). Fleshy algae of the genus *Cystoseira* prevailed in S1 and coralline algae together with the brown alga *Padina pavonica* in S2. The above stations differ mainly on exposure level [14,15].

Samplings were made in 2010 by scientists using SCUBA diving at approximately 8–10 m depth, and included a combination of visual census and random collection of specimens.

### 2.2. Population Density and Size Structure

The conventional distance sampling method, which models detection probability as a function of distance from a line or point transect and assumes the detection of all individuals at zero distance [16,17,18] was applied, in January 2010, to estimate the population density of *S. granularis*. Five (5) replicate 50-m line transects were surveyed in each station; replicate transects were randomly applied but desisted at least 10 m apart to avoid repetition. All living *S. granularis* individuals found along each random 50-m line transect were counted, and their perpendicular distance (d) from the line—under the assumption that d is negatively related with detection probability—within a 3 m belt astride to the line transect was also recorded with a tape-meter and inputted to Distance software (version 6, second release). Based on recorded distances, a detection function is fitted onto the data and used to estimate the detection probability, i.e., the proportion of non-detected individuals in the given area, and to assess the density of the studied population [16,17,18]. Distance measurements should be accurately recorded and the organism of interest should move slower that the observer to avoid biases. Although widely applied in terrestrial ecology, the method has been scarcely used for marine species other that mammals [19,20].

At each station, 20 sea urchin specimens were randomly collected at monthly intervals, from January to December 2010, to estimate the size structure of the studied *S. granularis* population. Concurrently, seawater temperature, salinity, pH, and dissolved oxygen were recorded with an autographic conductivity-temperature-depth sensor, CTD (SeaBird electronics, Washington, DC, USA).

In the laboratory, the sea urchin specimens were measured for test diameter (Dt) and height (Ht) using an electronic calliper (Mitutoyo Corporation, Takatsu Ward, Japan, 0.01 mm precision), and drained for 5-min on filter paper. Each specimen was, then, weighted for total weight (tW) using an electronic scale (0.001 g precision).

Analysis of variance, GLM ANOVA [21] was used to test for differences between the two sampling stations in the estimated biometric variables (Dt, Ht, tW). Prior to the analyses, data were tested for normality with the Anderson–Darling test, while the homogeneity of variance was tested with Cohran’s test. The Fisher LSD test was used for post-hoc comparisons when appropriate. ANOVAs were performed using the SPSS software package (IBM SPSS statistics version 19).

### 2.3. Reproductive Cycle

The monthly collected sea urchin specimens from both stations were further processed in the laboratory to assess the reproductive cycle of the species. More specifically, the sea urchins were dissected to remove and then, weigh the five gonads (Wg, 0.001 g precision). Weight measurements were used to calculate wet gonad-somatic index (GSI) as Wg/tW% to assess the reproductive condition of *S. granularis* [1]. GLM ANOVA was used to test for differences in GSI values between sampling stations (two-level random factor), months (12-level fixed factor), and of their interactions [21].

The specimens were sexed through the macroscopic examination of gonads or smear examination to calculate sex ratio (♂/♀), and the chi-square was apply to test for significant deviations from unity (1/1).

The middle portion of each individual’s gonads was fixed in formaldehyde, placed in cassettes and inputted in histokinette (Leica TP 1020, Leica Microsystems GmbH, Nussloch, Germany) for dehydration (immersion in ethanol solution of increasing concentrations), clearing (immersion in xylene solutions to replace ethanol with an organic dissolvent), and embedding in liquid paraffin wax. The gonadal tissue’ paraffin blocks were left for cooling (Leica EG 1150H, Leica Microsystems GmbH, Nussloch Germany) and then mounted on a microtome (Slee Mainz Cut 5062, SLEE medical GmbH, Mainz, Germany) for sectioning (5 μm sections). The sections were stained with the hematoxylin–eosin regressive staining procedure, covered with Canada balsam mounting medium, and observed under light microscopy (Axiostar plus Carl Zeiss optical microscope, Carl Zeiss Light Microscopy, Gottingen, Germany, connected with a ProgRes C10 digital camera, 10 × 100 magnification, JENOPTIC Optical Systems GmbH, Jena, Germany) for the histological examination of gonads. The different developmental stages of gametogenesis were assessed according to the 6-level (I = recovery, II = growing, III = premature, IV = mature, V = pre-spent, VI = spent) Byrne’s classification scale [22], for both female and male urchins, photographed and processed through image analysis (ProgRes Capture Pro 2.1 software). This scale, originally developed to assess the sea urchin’s *P. lividus* gametogenesis, has been also successfully applied in *S. granularis* [5].

## 3. Results

### 3.1. Environmental Parameters

The recorded environmental parameters generally showed similar values between the two sampling stations (Table 1). Sea-bottom temperature at 8 m depth varied annually from 13.28 to 27.67 and from 13.17 to 27.12 in S1 and S2, respectively, following the seasonal pattern of atmospheric warming in the study area. Salinity decreased in summer and autumn in both stations, whereas between-month differences are also observed. Dissolved oxygen varied temporally between sites, being generally increased at the exposed S2 station, while pH varied around 8.28 with very small differences between stations and months.

### 3.2. Population Density and Size Structure

Overall 76 sea urchins were detected by visual census along the five replicate line-transects (41 observations in S1 and 35 in S2). Mean population density (Table 2) over both stations was 5 individuals/100 m^2^ according to original field data (mean number of sea urchins observed and counted per transect); when these data are inputted to Distance software, the relevant value increased to 8.5 individuals/100 m^2^. Population density differed between sampling stations (*p* < 0.05) being higher in S1 (5.5 or 9.2 individuals/100 m^2^) compared to S2 (4.5 or 7.8 individuals/100 m^2^).

In total 480 *S. granularis* specimens were processed; 240 from S1 and 240 from S2. At S1, mean size was 69.63 ± 7.87 mm and 45.03 ± 6.51 mm for test diameter and height, respectively. In S2, the relevant sizes were 66.56 ± 8.27 mm and 42.37 ± 8.13 mm for Dt and Ht. Mean weight was 152.463 ± 47.176 and 136.722 ± 61.581, in S1 and S2, respectively.

ANOVA results revealed significant between-stations differences in the estimated biometric variables (test diameter Dt: F = 17.25 *p* < 0.01, test height Ht: F = 15.68 *p* < 0.01, total weight tW: F = 9.89 *p* < 0.01); both size and biomass of *S. granularis* were much higher in S1 (Figure 3). On the contrary, non-significant differences were detected in the urchin’s biometry between sexes (ANOVA results, *p* > 0.01).

Size frequency analysis, constructed for Dt was unimodal in both stations; modal length was at 70–75 mm in S1 and slightly lower at 65–70 mm in S2 (Figure 4). Size frequency distributions temporally skewed (data not shown) either to smaller (e.g., July, August, September) or larger modal lengths (e.g., March, April).

### 3.3. Reproductive Cycle

Overall, 265 *S. granularis* specimens were females and 215 males (S1: 139♀ and 101♂, S2: 126♀ and 114♂ in S2). The sex ratio was significantly biased in favour of females in S1 (S1: ♂/♀ = 0.72, x^2^ = 6.017 *p* < 0.05), whereas no significant deviation from unity was detected in S2 (S2: ♂/♀ = 0.90 x^2^ = 0.043 *p* > 0.05), being about 1:1.233 over the entire sea urchin population studied, suggesting a slight prevalence of females in Pagasitikos Gulf.

GSI ranged from 0.316 (September 2010) to 3.264 (March 2010) with a mean value at 1.980 ± 1.301 at S1 and 2.442 ± 1.163 at S2. GSI showed significant differences between stations (F = 16.62 *p* < 0.01), with increased values in S2. The relevant seasonal differences were also significant (F = 12.91 *p* < 0.001). In both stations, increased GSI values were recorded from late winter to early spring, in March in particular, whereas a secondary rise was also observed in July (Figure 5). The seasonal GSI trend was negatively correlated with the annual sea-water temperature, especially in S1 (Sρ = −0.81 and −0.34, for S1 and S2, respectively), whereas non-significant correlations are found with the other measured environmental variables.

The pattern of ovarian growth was divided into six stages (Figure 6). Recovering ovaries (Stage I) are filled with nutritive phagocytes together with some small, previtellogenic oocytes and few primary oocytes. At the next growing stage (Stage II), early vitellogenic oocytes appear. At the premature stage (Stage III) both small and large oocytes are present, although they are much less in number that small ones cohort. In the mature stage (Stage IV) the ovaries are filled with more closely packed and large ova (larger oocytes clearly prevailed in ova), whereas the germinal epithelium still contains oogonia. In the pre-spent stage (Stage V), small oocytes and nutritive phagocytes are no longer observable, whereas the ova further increased filling in-between gaps. However, some gaps may be present due to spawn, which will be further increase at the spent stage (Stage VI) where only few relict oocytes are present in an otherwise empty gonad.

The pattern of testis growth was also divided into six stages (Figure 7). Recovering testes (Stage I) contained only nutritive phagocytes. In growing testes (Stage II), spermatids and spermatocytes are observable; they will be further developed to spermatozoans in premature testes (Stage III), which will fill the testis in the mature stage (Stage IV). In pre-spent stage (Stage V) spermatozoans are densely packed in the testis, whereas in the spent stage (Stage VI) only few relict spermatozoans are observed in the testis’ lumen.

The relative frequencies of the maturity stages in female and male urchins, made for each station separately, showed a synchronized reproductive pattern between sexes (Figure 8). A clear annual reproductive cycle was evident, with a spawning peak in spring (March and April). Recovering gonads were observed in almost all months, but they clearly prevailed during summer. Although small differences between the two stations existed, such as the presence of immature males in February in S2, the overall annual reproductive pattern of the species was similar.

## 4. Discussion

*Sphaerechinus granularis* is a very common sea urchin inhabiting various sublittoral habitats and following specific depth-distributional patterns in different geographic sectors according to local environmental conditions [7]. In horizontally oriented mixed soft bottoms—consisting of seagrass meadows, muddy sands, and scattered boulders covered by various algal species, as the study area—the species thrives around 10 m. In the Mediterranean Sea, according to the very few relevant studies conducted for *S. granularis*, its density ranges from 1.8 to 315 individuals/100 m^−2^ in the NW (Spain) and SW (Algeria) sectors, respectively [9,23]. In the Aegean Sea, an overall density of 4.8 individuals/100 m^−2^ has been reported -a value very close to the one reported in the present study (5.0 individuals/100 m^−2^)—which, however, varied greatly between geographic sectors, being much lower in the oligotrophic southern part [7]. Such a pattern has been reported as well for *Paracentrotus lividus* and attributed to the reduced algal coverage of the rocky shore and accordingly to limited food supply [7]. Pagasitikos Gulf is characterized as meso-oligotrophic, with much increased nutrient inputs in its inner part, where S1 is located [13]. Therefore, the increased trophic status of the sheltered and innermost site may provide additional food supply to sustain a denser population.

All previous density reports for the species were based on original field data. However, by inputting these data to Density software—simulating the probability of the species to be overlooked– population density almost doubled. This suggest an underestimation of density values when based on direct field count data, with important implications for the management of the species. Accordingly, the conservation status of the sea urchin may have been favored as the population size may have been larger than originally judged. According to distance sampling density estimations, biases are typically due to the movement of the target species or random errors [18,24]. As *S. granularis* is a sedentary species, its motility patterns cannot bias density estimations (i.e., moving faster than the observer does), and so, relevant differences are most probably related with its cryptic behavior, i.e., active covering of the test with shells, pebbles, and algae [8]. The possibility of overlooking those well-hidden sea urchin individuals during underwater surveys, especially in vegetated bottoms, is increasing. Accordingly, the implementation of a distance sampling method that uses a detection function fitted to the set of records to estimate the proportion of missed animals and correct density estimates seems to be more accurate for predicting the abundance of such species.

Considering the size of the studied *S. granularis* population, significant differences were detected between the two sampling stations. According to the size spectra distributions, one mode at 70–75 mm was detected in S1, which is higher than the minimum landing size for the species (70 mm); in S2, the mode falls to 65–70 mm. Although the number of specimens was relatively low (20 specimens per month and station), this temporal trend, which conforms between the two stations, may be considered as indicative for the studied *S. granularis* population. GSI followed the opposite trend, with higher values in the exposed S2 station. The different environmental conditions—excluding temperature that showed very similar values between the two stations—may have favored the growth of *S. granularis* in the shelter and organically rich S1 station [14], as previously reported for other sea urchins [15]. *S. granularis* is a grazing species with typical herbivorous habits. It prefers feeding on decaying plant material and epiphytic algae, as well as on the rhizomes and roots of seagrasses or encrusting coralline algae [25]. It is considered as generalist feeder, consuming different food items according to their availability [26], and so, manifesting high phenotypic plasticity according to local environmental conditions, and especially food availability. The reduced hydrodynamics in S1 [13] resulting in increased nutrient supply and greater development of algal coverage (fleshy *Cystoseira*), may have forced resource allocation to somatic growth [10], in contrast to the greater investment to gonadal growth in S2. A similar population phenotypic response to different hydrodynamic conditions has been previously suggested for the species [27], as well as reduced somatic growth under food-limited conditions [28]. Mean gonadal weight of *S. granularis* was similar between the studied sites, and accordingly, the significantly reduced mean size of sea urchins in the exposed site may have caused the relevant rise in GSI. In March, where GSI picked in both sites, the collected sea urchins had reduced size and increased gonad weight in the exposed site, whereas the sex ratio was almost equal to unity in both sites. These results suggest the investment in gonadal growth under high hydrodynamics and probably less favored food supply. Differences in resource allocation have been reported for *S. granularis* [1], as well as for *P. lividus*. For the latter species, allocation of energy to somatic growth at the expense of reproduction has been reported from shelter sites, to explain size differences between sheltered and exposed sites [29]. Alternatively, illegal collection of larger urchins for local consumption from S2—S1 is restricted for fisheries—may explain the observed differences in the size-structure of the species.

*Sphaerechinus granularis* is a gonochoristic species with external fertilization. Its size at sexual maturity varies around 5 cm in test diameter [5]. The vast majority of the collected urchins were larger than 5 cm in diameter, and so, GSI and histological analysis are reliable to describe its reproductive biology in the study area. Females predominate in S1, whereas both sexes were equally represented in S2. A sex ratio of 3:2 or 3:3 has been reported for the species [1], whereas a bias in sex ratio balance in favor of females seems to be common in other sea urchin species [30,31].

Literature data on the reproductive biology of *S. granularis* are limited. In the Atlantic (Brittany), the species is reported to have an annual reproductive cycle with a single spawning season in spring [5,9]. In the Mediterranean Sea, two spawning events have been reported to occur along the French coasts (Nice), a minor one in late spring and a major one in late summer [11]. This contradicts to another study from the Alboran Sea that reports a single spawning episode in summer that may last until autumn, depending on prevalent environmental conditions at the studied locations, whereas mature gonads may be present all year round [1]. In the present study, a clear major spawning event in spring (March–April) was documented based on both the GSI and histology. GSI dropped a month earlier in S2 but remained in an overall higher state in this station, resulting probably from the smaller size of sea urchins, as previously discussed. The seasonal GSI pattern in the sheltered S1 conforms better to relevant trends from Brittany populations [9], and partly to western Mediterranean ones where the maxima are observed in summer [1]. However, according to GSI, a second minor spawning event in early autumn (September) may be inferred for the species, which, however was not confirmed by the histological examination of the gonads. Environmental factors—such as food availability, diet quality, and hydrodynamics—may affect GSI without initiating gametogenesis [32]. A second rise in GSI has been observed in both Atlantic [9] and Mediterranean [1] *S. granularis* populations, explained by nutritional factors that enhance gonad size, since histological data rejected the possibility of a second spawning event. Therefore, GSI results should be cautionary interpreted when describing the reproductive biology of sea urchins.

The results of the present study provide baseline knowledge on the biology of *S. granularis* in the Aegean Sea, and the eastern Mediterranean, where no previous information exists. The revealed plasticity in population parameters, even in closely located areas, may have serious implications for the species protection, by considering regulative measures based on local-scale knowledge (e.g., exact reproductive period, growth period) to viable manage its developing fishery. Moreover, as sea urchins’ metabolism is highly affected by climate change initiating increased energetic cost for the species survivorship and reproduction [33], recent comparative studies on the species biology are needed to predict future trends.

## Figures and Tables

**Figure 1 animals-10-01506-f001:**
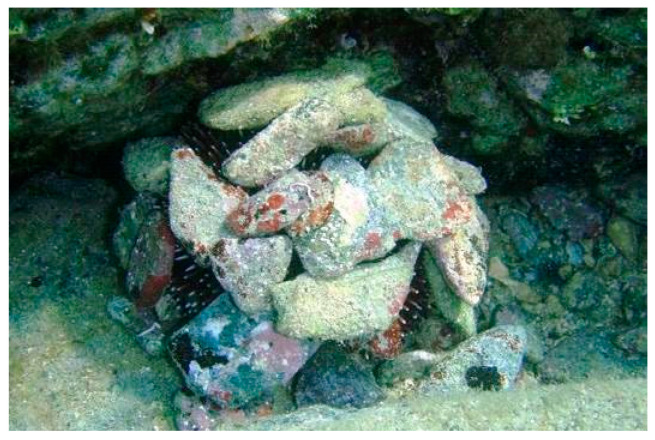
Covering behavior of the sea urchin *Sphaerechinus granularis*: underwater photography of a specimen in station 2, S2 (photo from Dr. Alexios Lolas).

**Figure 2 animals-10-01506-f002:**
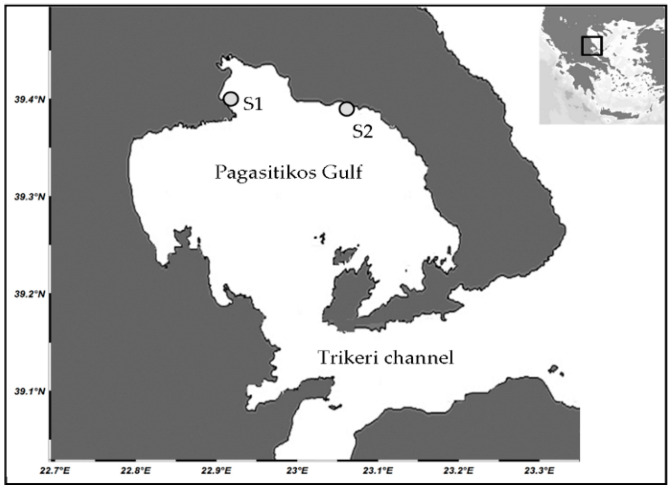
Map of the study area indicating the location of sampling stations (S1 and S2).

**Figure 3 animals-10-01506-f003:**
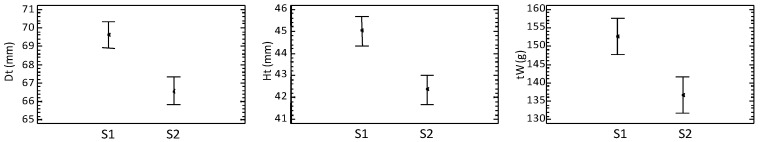
Mean size (test diameter Dt and height Ht in mm) and biomass (total weight tW in g) ± Fisher LSD of the sea urchin *Sphaerechinus granularis* in S1 and S2 sampling stations of Pagasitikos Gulf.

**Figure 4 animals-10-01506-f004:**
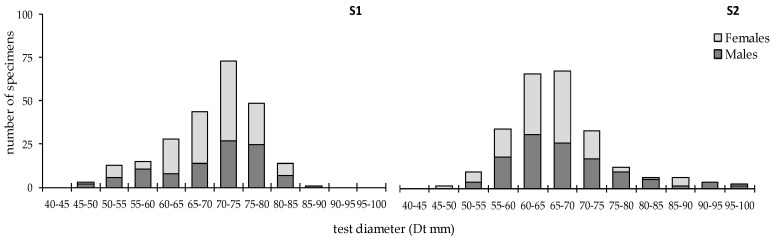
Size frequency histogram (Dt in mm) of the sea urchin *Sphaerechinus granularin* in S1 and S2 sampling stations of Pagasitikos Gulf.

**Figure 5 animals-10-01506-f005:**
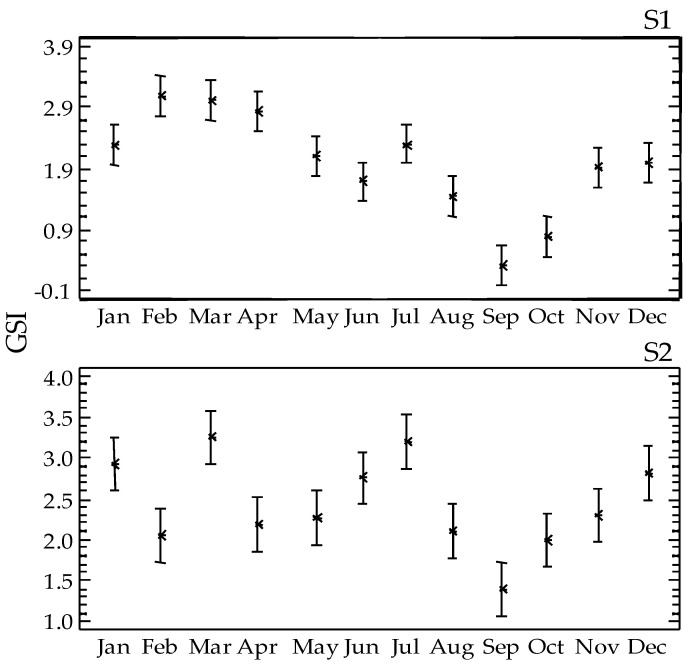
Temporal trend of the mean gonad-somatic index GSI ± Fisher LSD of *Sphaerechinus granularis* at the two sampling stations (S1, S2) in Pagasitikos Gulf.

**Figure 6 animals-10-01506-f006:**
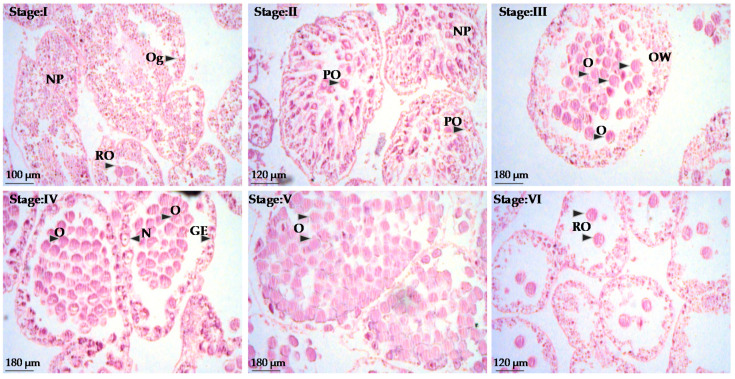
Histology of ovaries of *Sphaerechinus granularis*. NP = nutritive phagocytes, Og = oogenesis, RO = relict oocytes, PO = previtellogenic oocytes, O = oocytes, OW = ovarian wall, GE = germinal epithelium, N = nucleus, Stage I = recovery, Stage II = growing, Stage III= premature, Stage IV = mature, Stage V = pre-spent, Stage VI = spent.

**Figure 7 animals-10-01506-f007:**
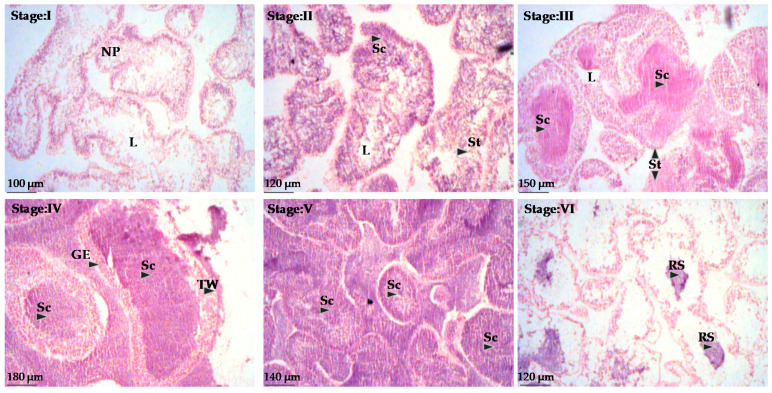
Histology of testes (right) of *Sphaerechinus granularis*. NP = nutritive phagocytes, L = lumen, Sc = spermatocytes, St = spermatids, GE = germinal epithethelium, TW = teste wall, RS = relict spermatozoa (stages are defined as in Figure 6).

**Figure 8 animals-10-01506-f008:**
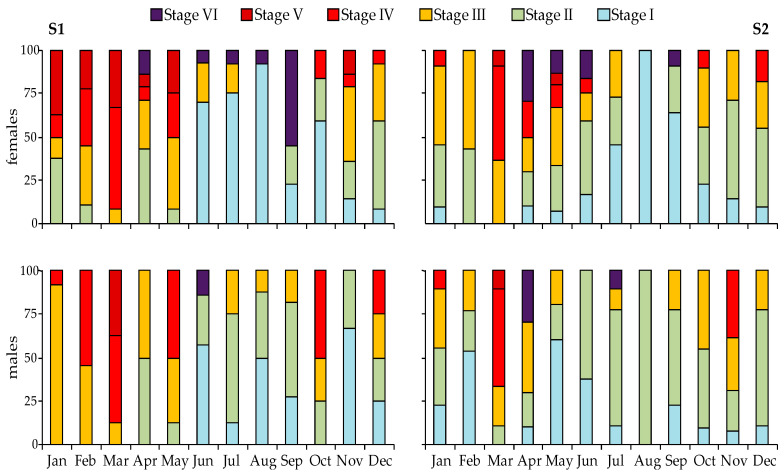
Temporal trend of the relative frequency of each gonadal developmental stage of female and male *Sphaerechinus granularis* individuals at the two sampling stations (S1, S2) in Pagasitikos Gulf (I = recovery, II = growing, III= premature, IV = mature, V = pre-spent, VI = spent).

**Table 1 animals-10-01506-t001:** Monthly values of the recorded environmental parameters, temperature T (°C), salinity S (psu) pH, and dissolved oxygen O_2_ (mg/L) in the sampling stations (S1 and S2) of Pagasitikos Gulf.

Month/Parameter	T		S		pH		O_2_	
	S1	S2	S1	S2	S1	S2	S1	S2
January	13.96	13.17	36.99	38.21	8.27	8.33	5.21	4.01
February	13.28	13.46	37.67	38.03	8.28	8.27	3.04	5.11
March	13.31	13.53	37.78	38.46	8.23	8.29	4.76	6.49
April	15.34	15.94	38.33	37.94	8.28	8.28	6.62	6.49
May	21.21	19.92	37.54	37.54	8.29	8.26	5.88	2.48
June	25.04	26.45	37.47	36.64	8.26	8.24	6.61	5.71
July	27.08	27.12	36.46	36.18	8.26	8.23	5.44	2.45
August	27.67	26.78	36.12	36.76	8.26	8.31	2.06	5.02
September	24.28	25.03	36.84	36.49	8.29	8.29	4.92	5.92
October	21.79	22.04	36.42	36.84	8.31	8.35	5.22	5.54
November	18.31	17.95	36.71	36.99	8.34	8.39	2.48	3.67
December	16.01	15.64	36.87	36.21	8.31	8.24	2.99	4.97

**Table 2 animals-10-01506-t002:** Population density of *Sphaerechinus granularis* in S1 and S2 sampling stations of Pagasitikos Gulf according to original field and Distance software data.

Station	Original Field Data Individuals/100 m^2^	Distance Software Data Individuals/100 m^2^
S1	5.5	9.2
S2	4.5	7.8
Mean value	5.0	8.5

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
