# Peer review of "Population Density, Size Structure, and Reproductive Cycle of the Comestible Sea Urchin Sphaerechinus granularis (Echinodermata: Echinoidea) in the Pagasitikos Gulf (Aegean Sea)"

_animals, 2020, doi:10.3390/ani10091506_

Round 1

Reviewer 1 Report

Dear authors,

I just came to read the revised version of your article with the title "Population density, size structure and reproductive cycle of the comestible sea urchin Sphaerechinus granularis (Echinodermata: Echinoidae) in Pagasitikos Gulf (Aegean Sea)" and you have made a great improvement of the document. I would accepted it as it is, but would advice you to reconsider again the issue with the early or delayed spawning events from S2 and S1 respectively. The fact that bigger urchins had smaller gonads (according to the GSI and not directly to the histological stage) does not in my opinion really imply a delayed spawning event and especially bigger urchins are more probably prone to support second gonadal cycles than small urchins do and even start earlier for this purpose. As S2 is exposed to uncontrolled fishery activities not present in S1, the GSI values should be treated with caution. I would add some small aspects in the discussion on the implication of your results for the species under climate change scenarios to support the validity of data being collected in 2010, which in my personal opinion are still valuable.

As the values calculated with the Distance software are highly different from the original data (40 % is quite big actually!) I would also suggest to include more details on which parameters you included as additional information for the calculation and also the number of observations analysed. Also some discussion on the implications of underestimated values should be included. I became interested in the software after reading your article and indeed, did not find references using it (even being released 10 years ago!), so I advice you to give as much details as possible to validate the use of such a tool also for future research. 

Here some references you could use to support your publication in terms of the software and parameters used:

  • Distance Sampling: Methods and Applications 2015, Buckland, S.T., Rexstad, E.A., Marques, T.A., Oedekoven, C.S. 
  • Distance software: design and analysis of distance sampling surveys for estimating population size 2010, https://doi.org/10.1111/j.1365-2664.2009.01737.x
  • Distance-Based Sampling of Software Configuration Spaces. 2019 Kaltenecker et al, https://doi.org/10.1109/ICSE.2019.00112 
  • Evaluation of the analysis of distance sampling data: a simulation study 2010, ROBERT EKBLOM, ORNIS SVECICA 20: 45–53
Kind regards,

Author Response

Thank you for your comments we considered them we added them here but we also send you the file tha contains them too.

Point 1: “I would advice you to reconsider again the issue with the early or delayed spawning events from S2 and S1 respectively. The fact that bigger urchins had smaller gonads (according to the GSI and not directly to the histological stage) does not in my opinion really imply a delayed spawning event and especially bigger urchins are more probably prone to support second gonadal cycles than small urchins do and even start earlier for this purpose. As S2 is exposed to uncontrolled fishery activities not present in S1, the GSI values should be treated with caution”.

The reviewer’s comment has been adopted; accordingly, we have deleted the “delayed spawning” parts from the discussion (L280; L283-284), which was originally referred to P. lividus.  

Point 2: “I would add some small aspects in the discussion on the implication of your results for the species under climate change scenarios to support the validity of data being collected in 2010, which in my personal opinion are still valuable”.

The reviewer’s comment has been adopted; accordingly, we have added this issue in the discussion section (L319-322).  

Point 3: “As the values calculated with the Distance software are highly different from the original data (40 % is quite big actually!) I would also suggest to include more details on which parameters you included as additional information for the calculation and also the number of observations analysed. Also some discussion on the implications of underestimated values should be included. I became interested in the software after reading your article and indeed, did not find references using it (even being released 10 years ago!), so I advice you to give as much details as possible to validate the use of such a tool also for future research. Here some references you could use to support your publication in terms of the software and parameters used: 1. Distance Sampling: Methods and Applications 2015, Buckland, S.T., Rexstad, E.A., Marques, T.A., Oedekoven, C.S.; 2. Distance software: design and analysis of distance sampling surveys for estimating population size 2010, https://doi.org/10.1111/j.1365-2664.2009.01737.x; 3. Distance-Based Sampling of Software Conguration Spaces. 2019 Kaltenecker et al, https://doi.org/10.1109/ICSE.2019.00112; 4. Evaluation of the analysis of distance sampling data: a simulation study 2010, ROBERT EKBLOM, ORNIS SVECICA 20: 45–53

The reviewer’s comment has been adopted. Accordingly, we have added more details on the applied method (L88-101, L151-153), we have discussed possible implications of underestimated density values (L244-247), and used additional references for distance-sampling (suggested references 1 and 2, as well as Katsanevakis 2009 and Thanopoulou et al. 2018 as the latter two refers on marine benthic species that are more relevant with our case).

Reviewer 2 Report

Dear Authors, 

thank you so much for the modifies addressed to the paper. 

It is now in a better form. 

I understand the difficulties in obtaining new data, but my doubts about reporting ten years old data still remain.

I leave the final decision about your paper in the hands of the Editors.

All the best 

Author Response

Thank you for your comments.

Reviewer 3 Report

Waiting for further studies.

Author Response

Thank you for your comments.

This manuscript is a resubmission of an earlier submission. The following is a list of the peer review reports and author responses from that submission.

Round 1

Reviewer 1 Report

Dear authors,

I gladly reviewed your article with the title "Population density, size structure and reproductive cycle of the comestible sea urchin Sphaerechinus granularis (Echinodermata: Echinoidae) in the Aegean Sea"and hereby I let you know my suggestions for improvement.

The article describes very important parameters to be taken into account for a correct management of fishery activities on this species and it is very relevant in order to avoid overfishing, as it was the case for P. lividus. You have used the Distance software for the evaluation of your data on population density, which is in my opinion a very innovative technique accounting for underestimations. The methodology you have used for the evaluation of gonadal development is correct and follows good scientific standards supported by recent bibliography.

Yet, in most publications, to the best of my knowledge, main raw data is presented for describing these population dynamic parameters. Therefore, I would suggest to add a table in the introduction accounting for published parameters for this species together with their environmental conditions (with environmental parameters relevant for triggering vitellogenesis, gonadal development and spawning events). This comparison of environmental parameters is of a great help for analysing patterns of energy allocation and will greatly improve the quality of your publication. Also more details on fishery activities is needed on the introduction, as this plays a great role on the evaluated parameters. If possible, add the respective with Distance Software calculated values (for population density), to have an idea of the underestimation ranges we are dealing with for other regions. This could also make your article better and distinctive from what have been already published for the studied sea urchin species.

The main text is well-written in a reader-friendly manner and I have only few remarks beside the specific comments:

  • the use of abbreviations should be consistent all over the document (see specific comments),
  • the given mode values should also be the same in all parts of the document (size structure is not consistent between Abstract, Results and Discussion),
  • I suggest to add scale bars in the nice pictures of ovaria and testes you presented.

I strongly recommend to discuss in more detail the issue on resource allocation and to go deeper into the analysis of possible sources of variation among the studied sites and mechanisms behind these responses (e.g. why sheltered environment leads to somatic growth instead of reproduction, the influence of feeding preferences on the selection of the habitat, territorial behaviour between different size cohorts and impact of this behaviour on reproduction). The comparison of environmental variables among different sites where this species is distributed may illustrate better what you have found on the studied area.

In general, I suggest minor revisions before publication and I hope my suggestions for improvement can help you. Please feel free to contact me again if clarification on the comments is needed.

Specific comments:

Abstract:

L17 and 26: What is mean with typical? The definition of typical covering behaviour should be shortly included, if not here (length constrains in abstract), at least in the introduction.

L19 and -L34: You defined two sampling locations and report 3 mode values for size structure in the abstract which do not match with what you present in results (L118-124, probably different units!!) and discussion (L200-205). Please correct. If the precision for this parameter is 0.01 (units missing), you should use in all referred parameters the same number of numbers after the comma (the same apply for GSI, which precision for mean values is not consistent)

L21: how many individuals were used for GSI calculations? If the same as for size structure, please rewrite the sentence for clear statement

L33: Have the Distance Software been already used in former publications? Could you please add some references for it?

L36-38: I find this last remark on spawning events for Atlantic and Mediterranean very interesting, I would suggest to go deeper in this issue in the introduction and discussion parts

Introduction:

L51-58: Please add more details on local fishery activities on the studied species and specifically on the sampled areas. Why were these areas chosen for the study? The description given in Materials and Methods match better the introduction part. I would suggest to go deeper on the implications of different algal coverage for the distribution of the different size cohorts

L62-68: Here a comparison table with information on the different regions where S granularis is distributed would be of great value

Materials and Methods:

L69: I would suggest to use subheading here as in Results. As Population density is so short, would be good to tackle this together with size structure (the same applies for results)

L86: Seawater temperature was recorded and is not presented in the text. It was not relevant for the results? This information is important for gonadal development as temperature and photoperiod are main triggers of reproduction cues. Therefore, I would suggest to add it, especially for comparisons of spawning events between different regions

L88-90: Abbreviations here do not match with further used in the text (e.g.L120). Please correct.

L89: missing units for the calliper precision. Please correct

L95: Chi-square test belongs to the part of the statistical analysis (L103-109)

Results:

L112-116: The values in parenthesis are original data and software-calculated data? I would suggest to make a small table in order to better present these values

L118-120: Abbreviations (see comment above)

L135-137: This statement match better the discussion part

L140: In the image capture I would suggest to delete “over the sampling months” as in this image there is no information on time

L141-148: Please correct the precision of the given GSI mean values. I would suggest to add the correlation analysis as supplementary materials if not tackled in more detail for the discussion as trigger parameters for reproduction cues

L160-164: Please add scale bars to the images on histology

Discussion:

L178-190: Please discuss in more detail the position of the studied sites with respect to the oligotrophic areas and why are there less individuals

L199-203: Please check the mode values referred (see comment above)

L209-216: Please discuss in more detail how the hydrodynamics of the site influence resource allocation and early- or delayed maturity. Include the influences of environmental factors triggering these processes, algal coverage, fishing activities on both sites. For me it is not clear why the authors inferred from Fig 8, that sheltered sites had delayed maturity. Maybe this explanation can be re-written in order to ease understanding of interpretation. Please check the size of the individuals of S2 in March, if they are similar to those in S1. If this is the case, they could be females left over by illegal fisheries and this factor would have a higher impact as expected and should be discussed deeper. What I understand from Fig 8 is that in S2 maturity stages are achieved between March and June (in S1 from December to May) and the spent stages remain for a shorter period of time from April to June, occasionally September (while in S1 this are present between April and September).

L229-235: What would you suggest to do in order to corroborate over histological examination the second spawning event that could be inferred from your results? From the GSI values it could be also possible to interpret that the smaller individuals had larger gonads in S2 (feeding/algal-coverage related effect?) I would suggest to check the higher values of GSI with respect to the size of the individuals and the months within this region. Moreover I would suggest to include the effect of temperature on the development of the gonads and discuss in more detail the seasonal difference in GSI, if it happens equally on both sites. Please discuss also in relation to GSI reported for the same season in other distribution areas.

L238-242: The last paragraph is mean as a kind of conclusion, but it sounds like introduction. I would suggest writing a real conclusion and stating how the obtained results could be used in order to protect this species in the future and what should be taken into account depending on the site.

Author Response

Please find the authors reply in the attached document. 

Reviewer 2 Report

Dear Authors,

Please find in the enclosed file my comments about the paper.

All the best

Author Response

(The authors gave the same response as above.)

Reviewer 3 Report

1. Abstract - vague description of the gonads, no reference to the annual cycle
2. Material and Methods - 36 gonads for cycle evaluation - not enough to evaluate the gonad development cycle
3. Methodology - correct
4. Result
5. Figure 6 - unfortunately the figures are not clear. The microscopic magnification is too low. In my opinion, it is impossible to determine individual cell types from these microscopic images.
6. In my opinion, the gonads were fixed too late, the cells shrunk what is visible in the photos. In fresh, properly fixed gonads, cells stick to each other, the nucleus is clearly visible.
Cell descriptions are also imprecise. There are oocytes in the prevention of PO, there are no vitellogenic oocytes marked in the figures. There is only a description of O = oocytes but it is not known what these oocytes are.
The same is true for the male gonads. Gonads were fixed too late, cells are not visible, magnification is too low. Higher magnification would also be of no use, as it is generally a "hag". It is intuitive to define a spawning and a frog gonad. On Stage V only marked with SD spermatids (?). How did the authors identify them? And where are the spermatozoa?
Pre-rubble gonads should be full of spermatozoids.
Actually, what we see in the figures of gonads gives some idea about the gonads, but it is difficult to distinguish them on the basis of these figures.
The authors base their analysis of the cycle on Byrne (1990). In this publication there are correct photos of female and male gonads, which show individual cells and it is possible to correctly interpret the stages of gonad development.
In the reviewed publication, the photos are much worse, although the image processing equipment is now more perfect.
Authors should also correctly sign the figures with the stages of development, as in Byrne (1990).
Figures should look like in the publications (examples):
Byrne, M. Annual reproductive cycles of the commercial sea urchin Paracentrotus lividus from an exposed 287 intertidal and a sheltered subtidal habitat on the west coast of Ireland. Mar. Biol., 1990, 104, 275–289.
Kirczuk et al. (2020) The Annual Reproductive Cycle of Rudd, Scardinius erythrophthalmus (Cyprinidae) from the Lower Oder River and Lake Dąbie, (NW Poland). Folia Biologica (Krakow), vol. 68 (2020), No 1 http://www.isez.pan.krakow.pl/en/folia-biologica.html https://doi.org/10.3409/fb_68-1.04
Labecka a., Domagala J. 2018. Continuous reproduction of Sinanodonta woodiana (Lea, 1824) females: an invasive mussel species in a female-biased population. Hydrobiology volume 810, pages57–76 (2018)
7. In the Discussion, the sexual cycle should be discussed in more detail, the differences between individuals from both positions and discussion with data from the literature
8. Publication possible after thorough corrections - placing clear figures of female and male gonads. This is the basis for the description of the results and their discussion.

Author Response

(The authors gave the same response as above.)

Reviewer 4 Report

Methods

Regarding the characterization of the stations S1 and S2, there is not any mention of the sea water temperature data collection in both of them, although the authors state that "the seasonal GSI trend was negatively correlated with the annual sea-water temperature" (section 3.3). What is the sea water temperature data source?

It would be useful to study the sea water temperature data along the year to assess any correlations between temperature differences in the two stations and the results obtained at least in relation to growth and gonadal development of the sampled specimens.

It would have been useful to apply the distance sampling method during the entire year of study in order to eventually assess any differences in seasonal population density of S. granularis. Is there any reason why the authors applied the distance sampling method only in December 2010?

Results

Pictures in figures 6 and 7 are lacking of magnification bars and their corresponding values. Moreover, the quality of the haematoxylin and eosin stained thick sections, especially in female gonads, is barely sufficient.

Regarding female, it is very difficult to identify clear differences between stage III and stage IV. The contents of mature oocytes should markedly differ from that of the pre-mature oocytes: the germinal vesicle of the mature eggs should show signs of breaking down or disappearing and their cytoplasm should be more homogenous and should stain very faintly with haematoxylin in respect to those of the pre-mature eggs.

It seems that the picture reported as stage V in figure 6 corresponds to an ovary with mature eggs, invalidating the pattern of ovarian growth used (reference n. 14).

Therefore, I wonder whether an incorrect classification of the female gonadal development stages may have led to evaluate a misguided temporal trend of the relative frequency of them (Figure 8).

Discussion

The authors state that the different environmental conditions between the two sampling stations can explain the significant differences detected as regards the size of the studied specimens and their GSI values.

If sea water temperature data are available, the authors can improve this section accordingly to further evaluation of interactions between the environmental conditions of the two stations and the biology of Sphaerechinus granularis.

Author Response

(The authors gave the same response as above.)

Reviewer 5 Report

The authors presented an interesting study about population density, size structure and reproductive biology of Sphaerechinus granularis, a sea urchin, in the Aegean Sea. I am not familiar with the species and histological analyses, so my capacity to review the manuscript is very limited. To the best of my knowledge, the paper is well written and the methods are sound. I have some minor comments to do:

L81: 'along five (5) replicate 50 m transects, in each station'. This sentence is a little confusing. I suggest to change to: 'A transect of 50 m was replicated 5 times in each station' or something similar.

L82: How was the 3 m distance measured? Were the transects marked? How did the authors guarantee that the same transect of 50 m was replicated?

L104: What do you mean with spatial-temporal differences? Which is the spatial level? Do you refer to the two stations? I doubt with two stations it is possible to measure spatial differences. In L125 I would not refer to spatial differences but to differences between stations.

L112: How many individuals were sighted in each transect?

L188: Do you refer to Distance software?

Author Response

(The authors gave the same response as above.)
